# A Modified Equivalent Factor Method Evaluation Model Based on Land Use Changes in Tianfu New Area

Jing Li [1,2], Jian Qiu [1,*], Majid Amani-Beni [1], Yuyang Wang [1], Mian Yang [1,3] and Juewen Chen [1]

1 School of Architecture, Southwest Jiaotong University, Chengdu 611756, China; 0720110013@mail.xhu.edu.cn (J.L.); majid@swjtu.edu.cn (M.A.-B.); ywang179@swjtu.edu.cn (Y.W.); yangmian@my.swjtu.edu.cn (M.Y.); chenjuewen2020@my.swjtu.edu.cn (J.C.)
2 School of Architecture and Civil Engineering, Xihua University, Chengdu 610039, China
3 Faculty of Art, Sichuan Tourism University, Chengdu 610100, China
* Correspondence: qiujian@home.swjtu.edu.cn

**Abstract:** Scientific understanding of urban ecosystem service value (*ESV*) is fundamental to building an urban ecological landscape pattern and improving urban environmental quality. The equivalent factor method (EFM) is widely used in evaluating *ESV* for natural ecosystems. In this study, using the EFM and sensitivity analysis, our research explored the space–time changes in land use and *ESV* during the planning and construction of Tianfu New Area from 2010 to 2020. This study selected correction factors from natural geography and social economy aspects, and established space–time correction models for standard equivalent coefficients as well as a comprehensive dynamic evaluation model for the ecosystem service value of specific urban areas. In terms of land use, the area of farmland decreased the most. The areas of construction land, grassland, and water bodies increased significantly. The reduced farmland was mainly converted into construction land, followed by grassland and water bodies. Other land use types had smaller changes. Due to the increased area of water bodies and their high-value coefficient per unit area, the urban ecosystem service value showed an increasing trend. During the study period, the conversion of about 1% of land led to about a 0.25% change in the urban ecosystem service value. Farmland in 2010 and water body in 2020 are the most sensitive land factors for *ESV* in Tianfu New Area. The results might have important insights for urban ecological environment protection and improving ecosystem services during the construction of newly built urban areas.

**Keywords:** urban ecological space; urban expansion planning; ecological environment protection; equivalent factor method



## 1. Introduction

Ecosystem services (ESs) refer to all the products and services that human beings receive from the ecosystem, which are the material basis and basic conditions for human survival and development [1,2]. ESs include service types such as provisioning service, regulating service, cultural service, and supporting service [3]. For the evaluation of ecosystem service value (*ESV*), scholars have carried out a lot of research [4–6]. The evaluation methods are roughly divided into three paradigms: quantity evaluation method, energy evaluation method, and value evaluation method [7,8]. The quantity evaluation method is based on the material flow in the ecological process and evaluates ESs from the perspective of material transformation [9]. The energy evaluation method uses the measurement standard of "solar energy" to quantitatively analyze ESs, which regards all ESs as the conversion of the ecosystem from solar energy. In this method, the evaluation index is too simplified, which may lead to inaccurate evaluation results [10,11]. The value evaluation method is the most widely used ESs evaluation method so far. Relying on ecological economics and environmental economics, the value evaluation method regards

ESs as valuable goods. It directly reflects the total value and scarcity level of ESs from the perspective of monetary value [7].

At present, the value evaluation method is mainly divided into two categories: the ecological modeling method (EMM) and the equivalent factor method (EFM) [12,13]. To implement EMM, it requires multiple data sources with a large number of inputting parameters, and hugely human and material resources. EMM contains a variety of complex calculation equations [13], and the evaluation process is complicated. EFM can be calculated based on the "equivalent value" and the corresponding area [14]. Compared with EMM, EFM is a widely used *ESV* evaluation method with fewer data demand and strong operability, which is intuitive [15]. Costanza et al. [1] clarified the evaluation principles, methods, and basic steps of EFM, estimated the incremental or marginal value of global ESs, laid the foundation of EFM evaluation, and greatly promoted the research progress of *ESV* scientific evaluation. Gaodi Xie et al. [5,12] localized the research results of Costanza et al. and established the equivalent value per unit area of ESs in China. The equivalent *ESV* coefficients have become the research basis for the revision of EFM and *ESV* evaluation by Chinese scholars. It is widely used in *ESV* evaluation of Chinese ecosystems. The research on *ESV* evaluation in China is mainly based on the direct citation of the value coefficient of Gaodi Xie [5,12]. For example, Zhang et al. took Zhongmou County as an example to study the changes in *ESV* in collaborative urbanization areas and found that the total *ESV* increased by CNY 10.05 million [16]. Some scholars have uniformly revised the value coefficient given by Gaodi Xie in one aspect, such as natural geography or social economy. For instance, Wang et al. used the average value of the biomass to modify the equivalent factor for ESs of each land use and land cover (LULC). They found that the total *ESV* improved remarkably by CNY 108.89 billion from 2001 to 2020 in the Yungui Plateau of China [17]. The equivalent value per unit area of ES in China [5,12] expresses the static value equivalent of ESs at a national scale and a specific time, ignoring the spatial heterogeneity and the dynamic changes over time caused by regional differences in biomass within the same land use type. In addition, Gaodi Xie's research objects mainly focus on natural ecological space, and *ESV* is mainly affected by natural geographical factors. According to the practical needs of the planning and construction of Tianfu New Area, Li and Qiu proposed value correction models based on systematic literature review and expert interviews. They determined five natural geographical and three socio-economic correction coefficients, the value correction in time dynamics and spatial heterogeneity of the equivalent *ESV* coefficients proposed by Gaodi Xie [5,12], and the comprehensive dynamic evaluation model of urban ecological space service value under the prominent influence of both natural geography and social economy [18].

The world is experiencing an unprecedented urbanization process characterized by the concentration of population, economy, technology, and other resources and the expansion of urban construction areas [19–21]. Urbanization is often accompanied by changes in LULC. Due to significant differences in the functions of ESs among different LULC types, changes in LULC directly affect urban *ESV*. Rahman et al. analyzed the impact of LULC changes on urban *ESV* in Dhaka, Bangladesh, from 1990 to 2020 and found that LULC changes are one of the important driving forces of ESs [22]. At present, scholars have begun to explore the driving factors of *ESV* changes caused by urbanization [23], and revealed the complex interaction and feedback relationship between urbanization and *ESV*. Some studies have shown a negative correlation between ESs and urbanization [24], mainly due to the rapid urbanization process, where more natural ecosystems are shifting towards impermeable surfaces, leading to a sharp decline in ESs supply. However, some research findings are exactly the opposite: *ESV* will increase in the process of urbanization. For example, Zhou et al. found that between 1996 and 2014, the *ESV* in Beijing–Tianjin–Hebei improved [25]. In addition, when using different indicators to measure the degree of urbanization, the relationship between *ESV* and urbanization may vary. Some studies have shown that when using population and economy to measure urbanization, *ESV* exhibits an

anti "U" relationship with urbanization [26,27]. However, when using land development, a negative correlation was found [28,29].

It is a long-standing challenge to balance conservation and development in the process of urban construction to maintain the urban ESs [8]. In this context, quantifying the *ESV* of urban ecological space (hereinafter referred to as "urban *ESV*"), studying the relationship between urban LULC changes and *ESV*, and mastering its change rules are the theoretical basis for carrying out urban ecological space performance evaluation, pattern optimization, and other work [15]. The scientific evaluation of *ESV* has important guiding significance for the formulation of effective urban management policies [30]. It can promote the improvement of urban space and living environment quality. This is conducive to the transformation of cities into green sustainable development. He et al. analyzed the space–time changes in land use and *ESV* in the Aral Sea Basin from 1993 to 2018 and found that *ESV* is closely related to land use, and effective land use policies can promote the sustainability of the ecosystem [31]. Ding et al. analyzed the land use structure and *ESV* change in Taiyuan City from 2003 to 2018, focusing on the response of *ESV* to the space–time evolution structure of land use in heavy industrial cities [32].

In November 2010, the Ministry of Housing and Urban-Rural Development of the People's Republic of China approved the "Coordinated Development Planning of Chengdu-Chongqing Urban Agglomeration", which proposed the construction of Tianfu New Area [33]. In October 2014, Tianfu New Area was officially approved by the State Council as a national new area (State Letter [2014] No.133) [34]. At this point, the construction of Tianfu New Area has been officially incorporated into the national development strategy. In February 2018, the concept of "Park City" was first proposed in Tianfu New Area [35]. In February 2022, the State Council authorized the approval of Chengdu to build a Park City demonstration area that practices the new development concept (State Letter [2022] No.10) [36]. It required that the ecological value should be fully demonstrated throughout the whole process of urban development. Tianfu New Area is located on the southern edge of Chengdu Plain, which is an important node for the construction of "The Belt and Road (B&R)" and the development of the Yangtze River Economic Belt [35]. Under the macro background of ecological civilization in China, Tianfu New Area undertakes the strategic task of national major development and reform and opening up, shoulders the important responsibility of helping China's development of western areas, building the Yangtze River Economic Belt, and protecting the ecological barrier of the Yangtze River [37]. Since its inception, Tianfu New Area has adhered to the concept of ecological priority and green development, from planning site selection, urban scale, and land layout to urban management. Tianfu New Area adhered to avoiding fertile land in planning site selection. It controls the urban scale with the short-board factor of bearing capacity (water resource factor) [33]. It plans the non-construction land first, followed by the construction land, to form various urban clusters. The implementation of the plan strictly adheres to the red line of ecological and basic farmland protection, and strictly protects the existing ecological green wedge in the planning area, and adheres to 70.1% of the blue-green ecological spatial layout, so that the city can grow organically between clear rivers and green mountains [35]. As the "First Mentioned Place" and "First Experimental Demonstration District" of Park City, Tianfu New Area has produced significant social, economic, and ecological benefits, laying the foundation for Chengdu to be approved as the only Park City demonstration area in China. Our research focuses on the impact of LULC changes on urban *ESV* of the mega-city new area in the context of China's national "Park City". The research's significance is to provide a reference for *ESV* evaluation and ecological space planning in urbanized areas, as well as suggestions for urban ecological policy-making.

## 2. Materials and Methods

### 2.1. Study Area

Chengdu city, the capital of Sichuan province, is a strategic economic hub in southwest China. Tianfu New Area is a national-level urban newly developed area of the city. Tianfu

New Area is located on the southern edge of Chengdu Plain (Figure 1) and is located in the ecological protection area at the upper reaches of the Yangtze River. Its geographical location is between 30°13′38″ and 30°40′23″ N latitude and 103°47′59″ and 104°15′34″ E longitude [38]. Tianfu New Area includes the Gaoxin Area, Shuangliu Area, Xinjin Area, Zhiguan Area, Longquan Area, and Jianyang Area within the jurisdiction of Chengdu city, Pengshan Area, and Renshou Area of Meishan city. Its total planning area is 1578 km² [38].

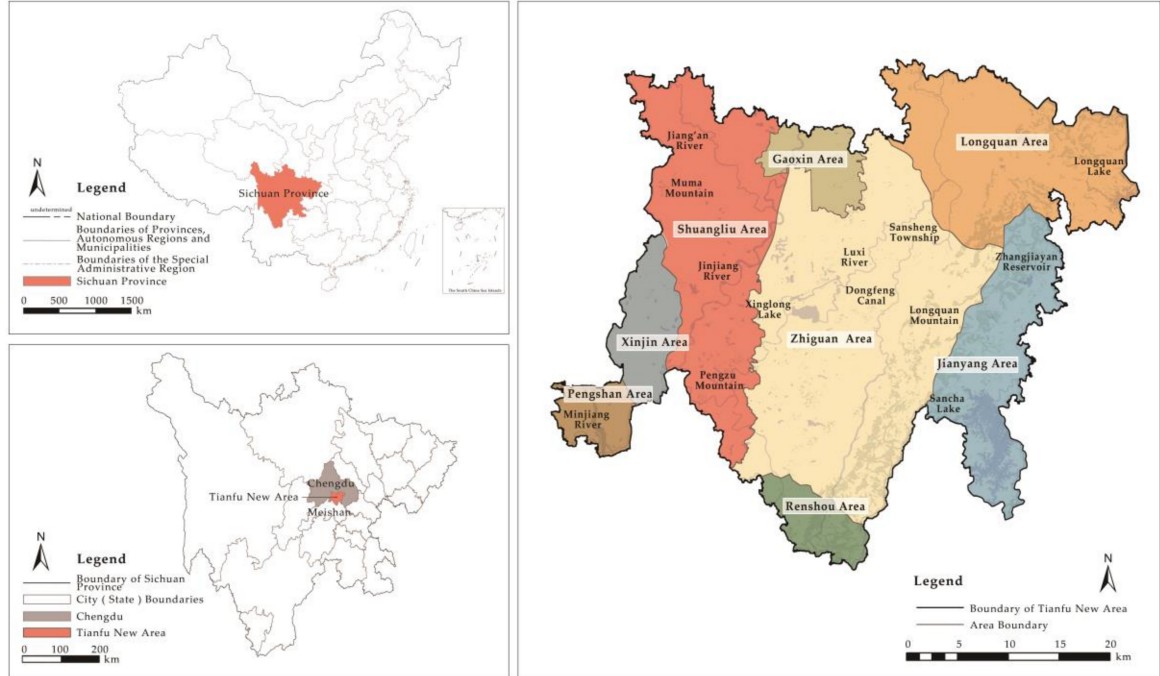

**Figure 1.** Area of study.

Chengdu Plain is surrounded by mountains, with four distinct seasons, mild climate and abundant rainfall, and is a temperate subtropical monsoon climate zone [39]. In most areas, the average annual temperature is 16.3 °C, precipitation is 855.8 mm, wind speed is 1.2 m/s, and the dominant wind is northeasterly. There are various landforms, mainly shallow hills. The areas of hills, mountains, and plains account for about 65%, 20%, and 15%, respectively [38]. The elevation in the area is between 350 and 1050 m. In general, the area is high in the east and low in the southwest, and the relatively high area is concentrated in Longquan Mountain.

### 2.2. Data Sources

This study employs multi-source data, including land use data, geospatial feature data, and socio-economic statistics of China and Tianfu New Area at two spatial scales in 2010 and 2020. The soil erosion data of China and Tianfu New Area in 2010 and 2020 were specially developed by the Fine Resolution Mapping of Mountain Environment (FRMM) project team. Specifically, it includes the following data: (1) Land use data of Tianfu New Area is from http://www.globallandcover.com (accessed on 6 May 2022) [40]. (2) Net Primary Productivity (NPP), precipitation, and national land use data are from the website "https://www.resdc.cn"(accessed on 18 May 2022) [41]. (3) Soil erosion data are from FRMM. (4) Road network data are from "Annual Report on Road Network Density in Major Chinese Cities". (5) The social and economic statistics data are from the national and local statistical yearbooks, and the net profit data of agricultural products are from the "National Farm Product Cost-Benefit Survey".

### 2.3. Methods

#### 2.3.1. Land Use Dynamic Index

To analyze the change of LULC in Tianfu New Area during the study period, we use the land use dynamic index to understand the change of LULC over time [22]. The calculation method is shown in Equation (1):

$$K = \frac{A_j - A_i}{A_i} \times \frac{1}{T} \times 100\% \tag{1}$$

where $K$ refers to the land use dynamic index of a single LULC type, $A_i$ and $A_j$ refer to the initial and final area of a specific LULC type, respectively, and $T$ refers to the research period.

#### 2.3.2. Calculation of *ESV*

According to Li and Qiu's method [18], the adjustment framework of the space–time correction model of standard equivalent coefficients in natural geography and social economy used in this study are as follows (Figure 2).

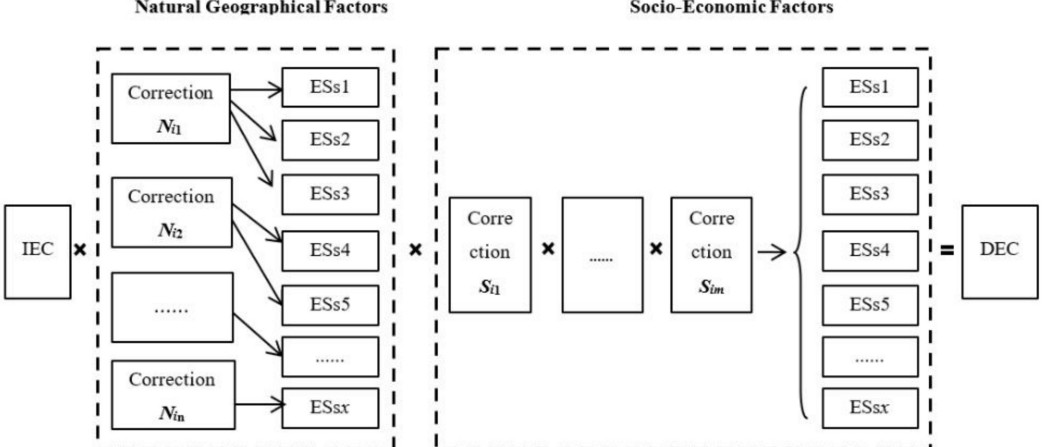

**Figure 2.** The adjustment framework of the space–time correction model [18]. IEC—the initial equivalent coefficients proposed by Gaodi Xie [12]; DEC—dynamic equivalent coefficients after natural geographical and socio-economic correction.

Space–Time Correction of Natural Geographical Factors

- Space–Time Correction of NPP ($N_{i1}$)

NPP was used to correct the space–time heterogeneity of food production, raw materials, gas regulation, climate regulation, waste treatment, and nutrient cycling in farmland, woodland, grassland, wetland, and desert. The equation is as follows:

$$N_{i1} = N_i / N \tag{2}$$

where $N_i$ refers to the annual average NPP in area $i$, and $N$ refers to the national annual average NPP.

- Space–Time Correction of Precipitation ($N_{i2}$)

Precipitation was used to correct the space–time heterogeneity of water supply and water regulation in farmland, woodland, grassland, wetland, desert, and water body. The equation is as follows:

$$N_{i2} = P_i / P \tag{3}$$

where $P_i$ refers to the annual average precipitation per unit area in area $i$, and $P$ refers to the national annual average precipitation per unit area.

- Space–Time Correction of Soil Conservation ($N_{i3}$)

Average soil erosion intensity was used to correct the space–time heterogeneity of soil conservation in farmland, woodland, grassland, wetland, and desert. The equation is as follows:

$$N_{i3} = E/E_i \qquad (4)$$

where $E_i$ refers to the annual average soil erosion intensity in area $i$, and $E$ refers to the national annual average soil erosion intensity.

- Space–Time Correction of Biodiversity ($N_{i4}$)

Average resistance of land type was used to correct the space–time heterogeneity of biodiversity in farmland, woodland, grassland, wetland, desert, and water body. The equation is as follows:

$$N_{i4} = B/B_i \qquad (5)$$

where $B_i$ refers to the annual average resistance of land use type in area $i$, and $B$ refers to the national annual average resistance of land use type.

- Space–Time Correction of Landscape Accessibility ($N_{i5}$)

$N_{i5}$ was used to correct the space–time heterogeneity of aesthetic landscape in farmland, woodland, grassland, wetland, desert, and water body and the road network density is used to express national and regional landscape accessibility. The equation is as follows:

$$N_{i5} = A_i/A \qquad (6)$$

where $A_i$ refers to the annual average road network density in area $i$, and $A$ refers to the national annual average road network density.

- Space–Time Correction Model of Natural Geographical Factors

Based on the above five corrections, the space–time correction model of natural geographical factors is calculated by Equation (7):

$$V_{ij\mathrm{cm}} = N_{it} \times V_t (t = 1, 2, 3, 4, 5) \qquad (7)$$

where $V_{ij\mathrm{cm}}$ refers to the equivalent coefficient of the ecosystem $j$ ES $c$ in area $i$ after space–time correction of natural geographical factors. $N_{it}$ refers to the $t$-type space–time correction in area $i$ (Equations (2)–(6) for $N_{i1}$, $N_{i2}$, $N_{i3}$, $N_{i4}$, and $N_{i5}$). $V_t$ refers to the equivalent coefficient proposed by Gaodi Xie et al. [12] before the space–time corrections. $V_1$ refers to food production, raw materials, gas regulation, climate regulation, waste treatment, nutrient cycling. $V_2$ refers to water supply and water regulation. $V_3$ refers to soil conservation. $V_4$ refers to biodiversity. $V_5$ refers to aesthetic landscape.

Space–Time Correction of Socio-Economic Factors

- Space–Time Correction of Resource Scarcity ($S_{i1}$)

The logarithm of population density was used to construct $S_{i1}$, and the space–time heterogeneity of 11 ESs in 6 ecosystems was corrected. The equation is as follows:

$$S_{i1} = \log R_i / \log R \qquad (8)$$

where $R_i$ refers to the average population density in area $i$, and $R$ refers to the national average population density.

- Space–Time Correction of Economic Development ($S_{i2}$)

The per capita GDP was used to construct $S_{i2}$, and the space–time heterogeneity of 11 ESs in 6 ecosystems was corrected. The equation is as follows:

$$S_{i2} = G_i / G \qquad (9)$$

where $G_i$ refers to the per capita GDP in area $i$, and $G$ refers to the national per capita GDP.

- Space–Time Correction of Social Development ($S_{i3}$)

The per capita general public budget expenditure was used to construct $S_{i3}$, and the space–time heterogeneity of 11 ESs in 6 ecosystems was corrected. The equation is as follows:

$$S_{i3} = F_i / F \tag{10}$$

where $F_i$ refers to the per capita general public budget expenditure in area $i$, and $F$ refers to the national per capita general public budget expenditure.

- Space–Time Correction Model of Socio-Economic Factors

Based on the above three corrections, the space–time correction model of socio-economic factors is calculated by Equation (11):

$$V_{ijcf} = S_{i1} \times S_{i2} \times S_{i2} \times V_{ijcm} \tag{11}$$

In the equation, $V_{ijcf}$ refers to the final equivalent coefficient of the ecosystem $j$ ES $c$ in area $i$ after space–time correction of socio-economic factors. The meanings of $S_{i1}$, $S_{i2}$, and $S_{i3}$ are shown in Equations (8)–(10). The meaning of $V_{ijcm}$ is shown in Equation (7).

Economic Price of One Standard Equivalent ES

Based on the research results of Gaodi Xie et al. [12], the equation is as follows:

$$D = S_r \times F_r + S_w \times F_w + S_c \times F_c \tag{12}$$

In the equation, $D$ refers to the economic price of one standard equivalent ES. $Sr$, $Sw$, and $Sc$ refer to the percentage of the sown area of rice, wheat, and maize in the total sown area of the three crops, respectively. $Fr$, $Fw$, and $Fc$ represent the average profit per unit area of the three crops above.

Comprehensive Dynamic Evaluation of Urban Ecological Space Service Value

According to the above space–time correction models from natural geography and socio-economy, namely Equations (2)–(12), the comprehensive dynamic evaluation model of urban ecological space service value can be obtained. The equation is as follows:

$$ESV_i = \sum_{c=1}^{m} \sum_{j=1}^{n} D \times V_{ijcf} \times A_j (c = 1, 2, \ldots, m; j = 1, 2, \ldots, n) \tag{13}$$

where $ESV_i$ refers to the total economic value of dynamic ESs in study area $i$. The meaning of $D$ is shown in Equation (12). $V_{ijcf}$ is shown in Equation (11). $A_j$ refers to the area of ecosystem $j$. $c$ refers to 11 types of ESs, such as food production, etc. $j$ refers to ecosystems, such as farmland, etc.

### 2.3.3. Annual *ESV* Change Rate

In order to understand the annual *ESV* change trend of a single LULC type, we calculated the *ESV* of each LULC type in 2010 and 2020, and used Equation (14) to estimate the annual *ESV* change rate of a single LULC type [22]:

$$ESV_{cr} = \frac{ESV_j - ESV_i}{ESV_i} \times \frac{1}{T} \times 100\% \tag{14}$$

where $ESV_{cr}$ refers to the annual *ESV* change rate of a single LULC type. $ESV_i$ and $ESV_j$ refer to the initial and final *ESV* of this LULC type, and $T$ refers to the research period.

2.3.4. Elasticity of *ESV* Due to LULC Changes

Elasticity describes the response of one variable to the change of another. In order to understand how the total amount of *ESV* varies with the change of LULC, we calculate the percentage change of *ESV* relative to the percentage change of LULC, which is used to reflect the elasticity of *ESV* [22]. The calculation method is shown in Equations (15) and (16):

$$EEL = \left| \frac{\frac{ESV_j - ESV_i}{ESV_i} \times \frac{1}{T} \times 100\%}{LTP} \right| \tag{15}$$

$$LTP = \left| \frac{\sum_{n=1}^{n} \Delta LCA_i}{\sum_{n=1}^{n} LCA_i} \times \frac{1}{T} \times 100\% \right| \tag{16}$$

*EEL* refers to the elasticity of total *ESV* related to LULC change. $ESV_i$ and $ESV_j$ refer to the initial and final ESV. *T* refers to the research period. *LTP* refers to the percentage of land conversion. $\Delta LCA_i$ refers to the area change of type *i* LULC, and $LCA_i$ refers to the area of type *i* LULC.

2.3.5. Sensitivity Analysis

To determine the dependence of total *ESV* on the change of a specific LULC value coefficient, according to the relevant references [31,42], the coefficient of sensitivity *(CS)* was used to analyze the relationship between ESs value coefficient *(VC)* and *ESV* to verify the accuracy of the evaluation results. The *VC* of each LULC type was adjusted (±) 50% [15,43], then the corresponding *ESV* changes were calculated. The equation is as follows:

$$CS = \left| \frac{\left(ESV_j - ESV_i\right) / ESV_i}{\left(VC_{jk} - VC_{ik}\right) / VC_{ik}} \right| \tag{17}$$

By calculating *CS*, we can understand the total *ESV* change level caused by the change of the *VC* of a specific LULC type. If *CS* > 1, it shows that *ESV* is more flexible to *VC*, and the reliability of the results is low. If *CS* < 1, *ESV* lacks flexibility to *VC*, and the results are more reliable [15,44]. $ESV_i$ and $ESV_j$ refer to the *ESV* before and after correction. $VC_i$ and $VC_j$ refer to the *VC* before and after adjustment. *k* refers to LULC type. Sensitivity analysis is widely used to understand *ESV* changes caused by changes in *VC* [22,31,32].

## 3. Results

### 3.1. Space–Time Changes in Land Use

The land use data of Tianfu New Area are from http://www.globallandcover.com (accessed on 6 May 2022), and the coordinate system adopted is *WGS_1984UTM_zone_48N*. The interpretation results are shown in Figure 3.

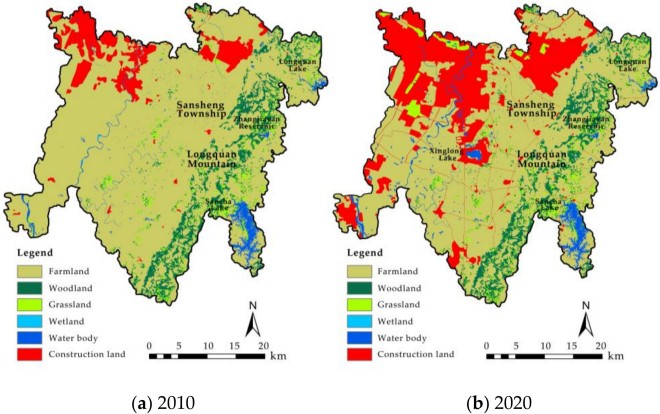

**Figure 3.** Land use status of Tianfu New Area in 2010 and 2020.

Using ArcGIS 10.7, the land use in Tianfu New Area in 2010 and 2020 were counted, and the land use dynamic index (K) was calculated to understand the annual change rate of different LULC types, as shown in Table 1.

**Table 1.** Land use changes of Tianfu New Area from 2010 to 2020.

| Land Use Types | 2010 | | 2020 | | 2010–2020 | |
|---|---|---|---|---|---|---|
| | Area (km$^2$) | Proportion (100%) | Area (km$^2$) | Proportion (100%) | Area Change (km$^2$) | K (100%) |
| Farmland | 1236.4 | 78.3% | 978.9 | 62.0% | −257.5 | −2.08% |
| Woodland | 124.5 | 7.9% | 122.9 | 7.8% | −1.6 | −0.13% |
| Grassland | 61.0 | 3.9% | 74.9 | 4.7% | 13.9 | 2.27% |
| Wetland | 0.3 | 0.02% | 0.5 | 0.03% | 0.2 | 6.64% |
| Water body | 36.5 | 2.3% | 48.0 | 3.0% | 11.5 | 3.16% |
| Construction land | 119.7 | 7.6% | 353.2 | 22.4% | 233.5 | 19.51% |

To understand the mutual transformation of specific LULC types during the research period, we created a transfer matrix using ArcGIS 10.7, as shown in Table 2.

**Table 2.** Land use transfer matrix of Tianfu New Area from 2010 to 2020 (unit: km$^2$).

| Types | Farmland | Woodland | Grassland | Wetland | Water Body | Construction Land |
|---|---|---|---|---|---|---|
| Farmland | 956.5 | 8.8 | 19.7 | 0.2 | 16.2 | 235.1 |
| Woodland | 9.5 | 102.0 | 11.6 | 0.02 | 0.2 | 1.2 |
| Grassland | 6.0 | 11.9 | 38.6 | 0.3 | 1.1 | 3.1 |
| Wetland | 0.06 | No wetland into woodland | No wetland into grassland | 0.02 | 0.2 | 0.01 |
| Water body | 4.5 | 0.2 | 0.5 | 0.04 | 29.2 | 2.0 |
| Construction land | 2.3 | 0.01 | 4.5 | No construction land into wetland | 1.0 | 111.8 |

Tables 1 and 2, and Figure 3 show the distribution and changes of different LULC types in the study area from 2010 to 2020. The land use in Tianfu New Area is mainly farmland, accounting for 78.3% in 2010 and 62.0% in 2020, mainly distributed in the central and western parts. Woodland and construction land follow, with woodland distributed in the Longquan Mountain area on the southeast side, and the construction land mainly distributed in the area on the north side connected with the main city. Grassland, water body, and wetland make up a small percentage. From 2010 to 2020, the whole land use changed obviously, and its changes showed that the land use types with the most significant change in the area were farmland and construction land. The farmland area decreased by 257.5 km$^2$. The construction land area increased by 233.5 km$^2$ and jumped to second place with 22.4% of the total area in 2020, obviously surpassing the woodland. From 2010 to 2020, more than 80% of the reduced farmland was converted into construction land. In addition to construction land, the conversion of farmland to grassland and farmland to water body were the main types of land conversion. The changes in other land use types were relatively flat compared with the above. Among them, the land use types with the increased area were grassland, water body, and wetland, which increased by 13.9 km$^2$, 11.5 km$^2$, and 0.2 km$^2$, respectively, and the woodland decreased by 1.6 km$^2$.

From the perspective of spatial distribution (Figures 1 and 3), the fastest-growing construction land is mainly concentrated in the north of the planning area, which is connected with the main urban area of Chengdu. In the process of urban construction, it has

the advantages of perfect municipal infrastructure and flat terrain and has become the priority area for development and construction. Among them, the urban construction land on the north side is separated by an ecological green wedge from Longquan Mountain to Sansheng Township, forming a cluster-like layout of urban construction land, retaining the urban ventilation corridor and preventing the urban adhesion development, which meets the requirements of the urban planning. The newly added urban construction land in the northwest is mainly concentrated in Shuangliu Area, Gaoxin Area, and Zhiguan Area near Xinglong Lake, while the northeast is mainly concentrated in Longquan area. In addition, some new urban construction land is added in Xinjin Area and Pengshan Area in the southwest and Renshou Area in the south.

### 3.2. Calculation of ESV

In this study, the total *ESV* of all land use types as well as all ESs types in Tianfu New Area in 2010 and 2020 were calculated according to Equation (13) (Tables 3 and 4, Figures 4 and 5).

**Table 3.** Summary of ecological space service value of Tianfu New Area in 2010 (unit: CNY 10,000).

| ESs | | Farmland | Wood Land | Grassland | Wetland | Water Body | Total of ESs | Proportion of ESs |
|---|---|---|---|---|---|---|---|---|
| Provisioning | FP | 98,476 | 2795 | 942 | 11 | 1421 | 103,645 | 11.9% |
| | RM | 22,125 | 6334 | 1456 | 11 | 410 | 30,336 | 3.5% |
| | WS | 114,528 | 3276 | 771 | 54 | 21,394 | −89,033 | −10.2% |
| Regulating | GR | 78,955 | 20,966 | 5010 | 40 | 1370 | 106,341 | 12.2% |
| | CR | 41,647 | 62,724 | 13,211 | 76 | 4071 | 121,729 | 14.0% |
| | WT | 12,581 | 17,778 | 4368 | 76 | 9871 | 44,674 | 5.1% |
| | WR | 131,447 | 30,925 | 9528 | 504 | 263,836 | 436,240 | 50.1% |
| Supporting | SC | 29,499 | 16,511 | 3918 | 31 | 1652 | 51,611 | 5.9% |
| | NC | 14,315 | 1922 | 471 | 4 | 128 | 16,840 | 1.9% |
| | BD | 7808 | 11,837 | 2826 | 85 | 3393 | 25,949 | 3.0% |
| Cultural | AL | 6507 | 9391 | 2270 | 92 | 4571 | 22,831 | 2.6% |
| Total of LULC | | 328,832 | 184,459 | 44,771 | 984 | 312,117 | 871,163 | 100% |
| Proportion of LULC | | 37.8% | 21.2% | 5.1% | 0.1% | 35.8% | 100% | -- |

FP—food production, RM—raw materials, WS—water supply, GR—gas regulation, CR—climate regulation, WT—waste treatment, WR—water regulation, SC—soil conservation, NC—nutrient cycling, BD—biodiversity, AL—aesthetic landscape.

**Table 4.** Summary of ecological space service value of Tianfu New Area in 2020 (unit: CNY 10,000).

| ESs | | Farmland | Woodland | Grassland | Wetland | Water Body | Total of ESs | Proportion of ESs |
|---|---|---|---|---|---|---|---|---|
| Provisioning | FP | 68,349 | 2415 | 1025 | 16 | 1752 | 73,557 | 7.8% |
| | RM | 15,800 | 5476 | 1576 | 15 | 505 | 23,372 | 2.5% |
| | WS | −108,190 | 3795 | 1130 | 107 | 33,593 | −69,565 | −7.4% |
| Regulating | GR | 54,954 | 18,154 | 5360 | 59 | 1685 | 80,212 | 8.5% |
| | CR | 29,194 | 54,376 | 14,241 | 111 | 5020 | 102,942 | 10.9% |
| | WT | 8587 | 15,438 | 4703 | 111 | 12,164 | 41,003 | 4.4% |
| | WR | 123,989 | 36,395 | 13,978 | 1002 | 414,240 | 589,604 | 62.6% |
| Supporting | SC | 22,325 | 15,438 | 4545 | 50 | 2039 | 44,397 | 4.7% |
| | NC | 9960 | 1682 | 525 | 6 | 152 | 12,325 | 1.3% |
| | BD | 5496 | 10,392 | 3074 | 125 | 3959 | 23,046 | 2.4% |
| Cultural | AL | 4808 | 8624 | 2601 | 143 | 5593 | 21,769 | 2.3% |
| Total of LULC | | 235,272 | 172,185 | 52,758 | 1745 | 480,702 | 942,662 | 100% |
| Proportion of LULC | | 24.9% | 18.3% | 5.6% | 0.2% | 51.0% | 100% | -- |

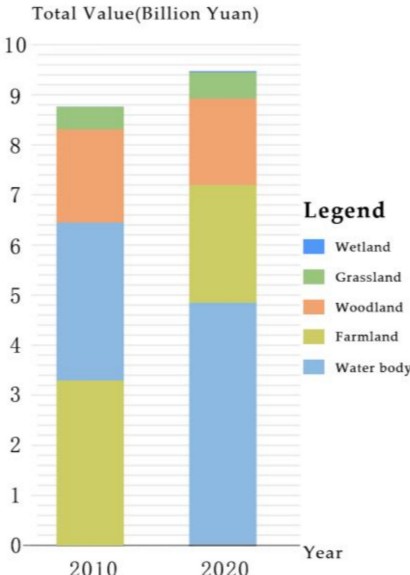

**Figure 4.** *ESV* composition of land use in Tianfu New Area in 2010 and 2020.

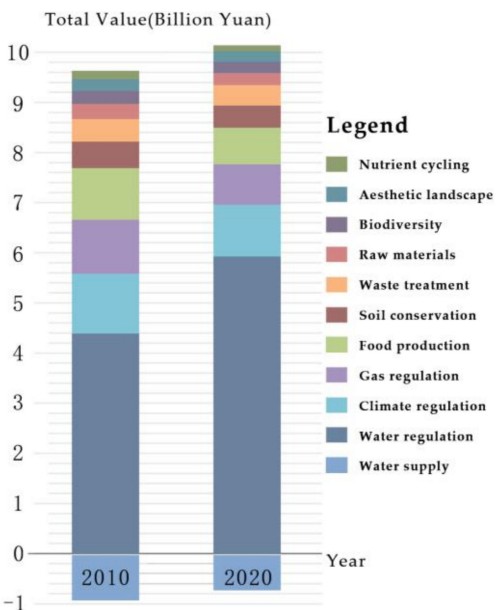

**Figure 5.** Value composition of ESs in Tianfu New Area in 2010 and 2020.

In 2010, the total *ESV* of Tianfu New Area was CNY 8711.63 million (Table 3), and in 2020 it was CNY 9426.62 million (Table 4). As for the value contribution of land use types (Figure 4, Tables 3 and 4), in 2010, 37.8% of the total *ESV* came from farmland, 35.8% from water body, and 21.2% from woodland. In 2020, 51.0% came from water body, 24.9% from farmland, and 18.3% from woodland and the sum of them accounted for more than 94% of total *ESV*, which were the main sources of total *ESV*. Over the 10 years, due to the increase of water body and the decrease of farmland in the area, the water body had contributed more *ESV* than farmland and became the first contributor to total *ESV* in 2020. Except for water body, farmland, and woodland, the contribution of grassland was higher than wetland in *ESV* in 2010 and 2020 because the wetland area was too small and its value was the lowest.

As for the value contribution of the secondary ESs types (Figure 5, Tables 3 and 4), the ranking in 2010 and 2020 is the same, from highest to lowest is water regulation > climate regulation > gas regulation > food production > soil conservation > waste treatment >

raw materials > biodiversity > aesthetic landscape > nutrient cycling > water supply. The top four secondary ESs accounted for 88.2% and 89.8% of the total *ESV* in 2010 and 2020, respectively, and were the main contributors to the total *ESV*.

As for the spatial distribution of *ESV* (Figure 6), high-value areas of *ESV* are concentrated in the vicinity of water body and woodland, such as Sancha Lake, Longquan Lake, Zhangjiayan Reservoir, and Longquan Mountain on the southeast side. Xinglong Lake, which was built during the construction process, became a high-value area in 2020. Low-value areas are mainly distributed in urban construction areas and some rural settlements in the north of Tianfu New Area. The median area is mainly distributed in the farmland area with low *ESV* per unit area in the northwest of Longquan Mountain, which is the most widely distributed area.

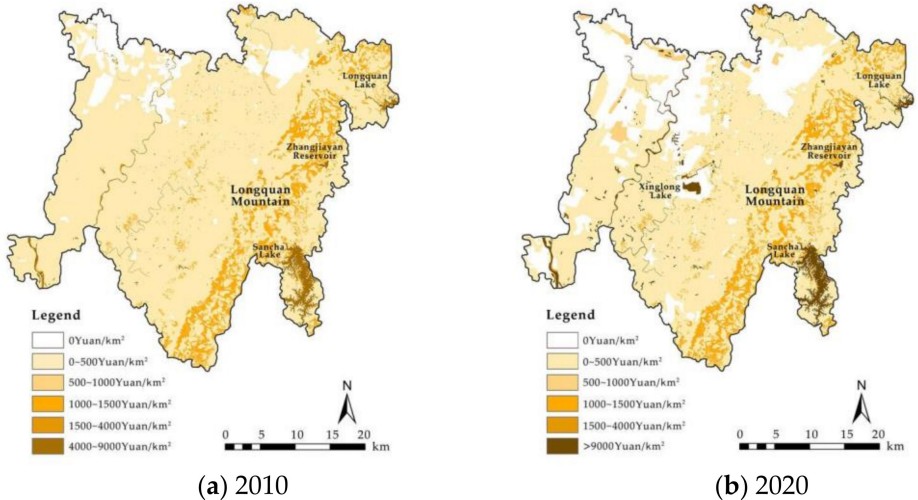

**(a)** 2010  **(b)** 2020

**Figure 6.** Spatial distribution of *ESV* per unit area of Tianfu New Area in 2010 and 2020.

### 3.3. Space–Time Changes of ESV

#### 3.3.1. Time Change of *ESV*

The total *ESV* of Tianfu New Area was CNY 8711.63 million in 2010 and CNY 9426.62 million in 2020, with an overall increase of CNY 714.99 million. To understand the annual *ESV* change rate, we calculated the $ESV_{cr}$ from 2010 to 2020 according to Equation (14). See Table 5 for details.

In terms of land use types, due to the reduction of the area of farmland and woodland, the *ESV* of farmland and woodland had shown a decreasing trend over the 10 years. The *ESV* reduction of farmland was CNY 935.6 million ($ESV_{cr}$ was −2.85%) and woodland was CNY 122.74 million ($ESV_{cr}$ was −0.67%). Since the value coefficient of water supply of farmland is negative, the water resources consumption of farmland has decreased due to the reduction of farmland area in the 10 years, so the water supply of farmland had shown an increasing trend, with an increase of CNY 63.38 million. Except for farmland and woodland, the other three types of land use had shown an increasing trend, with the increasing range of water body (CNY 1685.85 million, $ESV_{cr}$ was 5.4%) > grassland (CNY 79.87 million, $ESV_{cr}$ was 1.78%) > wetland (CNY 7.61 million, $ESV_{cr}$ was 7.73%). Among them, the *ESV* of water body had increased significantly and was the most important contributor to the positive growth of total *ESV*. The main reason is that the value coefficient of water body is much higher than other land use types, and the water body area in Tianfu New Area has increased substantially in the 10 years.

**Table 5.** Summary of ecological space service value changes of Tianfu New Area from 2010 to 2020.

| ESs | | Farmland | | Woodland | | Grassland | | Wetland | | Water Body | | Total Value of ESs | |
|---|---|---|---|---|---|---|---|---|---|---|---|---|---|
| | | Value (CNY 10,000) | Proportion (100%) | Value (CNY 10,000) | Proportion (100%) | Value (CNY 10,000) | Proportion (100%) | Value (CNY 10,000) | Proportion (100%) | Value (CNY 10,000) | Proportion (100%) | Total Value (CNY 10,000) | Proportion (100%) |
| Provisioning | FP | −30,127 | 32.2% | −380 | 3.1% | 83 | 1.0% | five | 0.7% | 331 | 0.2% | −30,088 | −42.1% |
| | RM | −6325 | 6.8% | −858 | 7.0% | 120 | 1.5% | four | 0.5% | 95 | 0.1% | −6964 | −9.7% |
| | WS | 6338 | −6.8% | 519 | −4.2% | 359 | 4.5% | 53 | 7.0% | 12,199 | 7.2% | 19,468 | 27.2% |
| Regulating | GR | −24,001 | 25.7% | −2812 | 22.9% | 350 | 4.4% | 19 | 2.5% | 315 | 0.2% | −26,129 | −36.5% |
| | CR | −12,453 | 13.3% | −8348 | 68.0% | 1030 | 12.9% | 35 | 4.6% | 949 | 0.6% | −18,787 | −26.3% |
| | WT | −3994 | 4.3% | −2340 | 19.1% | 335 | 4.2% | 35 | 4.6% | 2293 | 1.4% | −3671 | −5.1% |
| | WR | −7458 | 8.0% | 5470 | −44.6% | 4450 | 55.7% | 498 | 65.4% | 150,404 | 89.2% | 153,364 | 214.5% |
| Supporting | SC | −7174 | 7.7% | −1073 | 8.7% | 627 | 7.9% | 19 | 2.5% | 387 | 0.2% | −7214 | −10.1% |
| | NC | −4355 | 4.7% | −240 | 2.0% | 54 | 0.7% | 2 | 0.3% | 24 | 0% | −4515 | −6.3% |
| | BD | −2312 | 2.5% | −1445 | 11.8% | 248 | 3.1% | 40 | 5.3% | 566 | 0.3% | −2903 | −4.1% |
| Cultural | AL | −1699 | 1.8% | −767 | 6.2% | 331 | 4.1% | 51 | 6.7% | 1022 | 0.6% | −1062 | −1.5% |
| Total value of LULC | | −93,560 | 100% | −12,274 | 100% | 7987 | 100% | 761 | 100% | 168,585 | 100% | 71,499 | 100% |
| $ESV_{cr}$ | | —— | −2.85% | —— | −0.67% | —— | 1.78% | —— | 7.73% | —— | 5.4% | —— | 0.82% |

In terms of the secondary ESs types, nine kinds of services had shown a decreasing state over the 10 years, with the decreasing range of food production (CNY 300.88 million) > gas regulation (CNY 261.29 million) > climate regulation (CNY 187.87 million) > soil conservation (CNY 72.14 million) > raw materials (CNY 69.64 million) > nutrient cycling (CNY 45.15 million) > waste treatment (CNY 36.71 million) > biodiversity (CNY 29.03 million) > aesthetic landscape (CNY 10.62 million). The main reason is that the *ESV* per unit area of the nine kinds of services had been reduced to different degrees. In addition, farmland, which is the main contribution land use type for food production, and woodland, which is the main contributor to gas regulation, climate regulation, soil conservation, raw materials, nutrient cycling, and waste treatment, are all in decreasing states in areas. Two kinds of services are on the rise, and the increase range is water regulation (CNY 1533.64 million) > water supply (CNY 194.68 million). The main reason is that the *ESV* per unit area of the two kinds of services had increased greatly, and the main contribution land use types of water regulation and water supply–water body, wetland, and grassland had increased substantially, and the main consumption land use type of water supply–farmland had decreased conspicuously in area.

3.3.2. Spatial Change of *ESV*

As for the spatial change of *ESV* per unit area (Figure 7), from 2010 to 2020, the *ESV* per unit area of 95.7 km$^2$ of land in Tianfu New Area increased. It mainly consists of the former water body with an increased value per unit area, and the farmland converted into water body and grassland, and the grassland converted into woodland, the four-above account for 80.5% of the land area whose *ESV* per unit area increased. Among them, the land area with a lifting range of CNY 50–100 million/km$^2$ is 18.8 km$^2$. It is mainly composed of farmland converted into water body, accounting for 86.2%, with a value per unit area increase of CNY 97.5 million/km$^2$. This part is concentrated in Xinglong Lake Area. The land area with a lifting range of CNY 10–50 million/km$^2$ is 38.5 km$^2$. It is mainly composed of the former water body whose value per unit area has increased by CNY 14.6 million /km$^2$. It is distributed in Sancha Lake, Longquan Lake, and Zhangjiayan Reservoir in the east and Minjiang River in the west. The types with the promotion range of CNY 0–10 million/km$^2$ are mainly composed of grassland converted into woodland and farmland converted into grassland. The area of both accounts for 82.1% of this interval. It increased by CNY 6.7 million/km$^2$ and CNY 4.4 million/km$^2$, respectively. It is distributed in the green space built in the northern urban construction area and near Longquan Mountain.

The *ESV* per unit area of about 1373.9 km$^2$ in the study area is reduced. It is mainly composed of the former farmland, woodland with a slightly reduced value per unit area, and the farmland converted into construction land, which accounts for 94.4% of the land with reduced *ESV* per unit area. Among them, the land area with the reduction interval of *ESV* per unit area of CNY 0–1 million/km$^2$ is 1100.1 km$^2$, which is the former farmland, woodland, and grassland with unchanged land use and slightly reduced value per unit area. The areas are 959.5 km$^2$, 102.0 km$^2$, and 38.6 km$^2$, respectively. It is distributed in Longquan Mountain in the southeast, the ecological green wedge from Longquan Mountain to Sansheng Township in the north, and Shuangliu Area and Xinjin Area in the southwest. The area with a reduced range of CNY 1–10 million/km$^2$ is 255.8 km$^2$. It is mainly composed of the former farmland converted into construction land, accounting for 91.9%, and the value per unit area is reduced by CNY 2.7 million/km$^2$. It is distributed in the area connected with the main city on the north side of the study area, the area around Xinglong Lake in the middle, Xinjin Area in the southwest, and Renshou Area in the south.

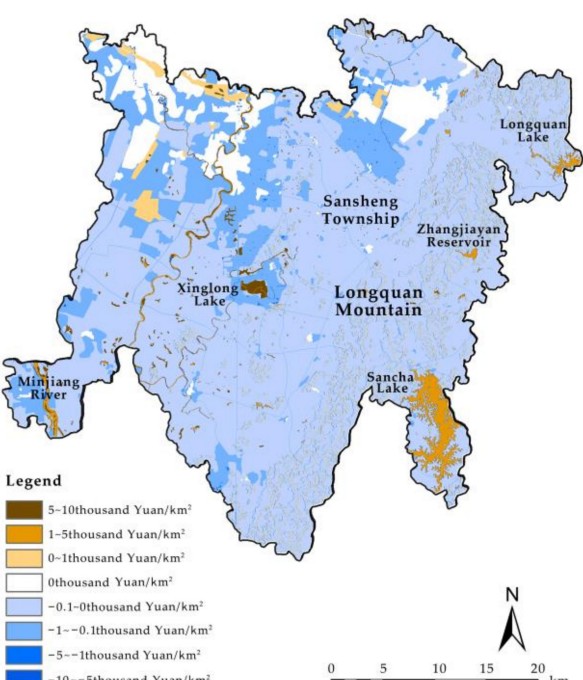

**Figure 7.** Spatial distribution of *ESV* changes per unit area in Tianfu New Area from 2010 to 2020.

### 3.4. Impact of Land Use Changes on ESV

To understand the influence of LULC changes on *ESV*, we calculated the elasticity of total *ESV* related to LULC changes (*EEL*) according to Equations (15) and (16). According to the calculation, the *EEL* of Tianfu New Area is 0.25 from 2010 to 2020. It means that 1% change of LULC led to an average change of 0.25% of total *ESV*. The higher the elasticity, the more sensitive *ESV* is to LULC changes. It is found that *ESV* is sensitive to LULC changes in Tianfu New Area from 2010 to 2020.

### 3.5. Sensitivity Analysis

*CS* can be used to verify whether the calculated results of *ESV* coefficients are reliable or not. To ensure the evaluated *ESV* results are reliable, the *CS* should be less than 1 [15]. Through the calculation, the *CS* of each land use is less than 1 (Table 6), and farmland > water body > woodland > grassland > wetland in 2010, water body > farmland > woodland > grassland > wetland in 2020. It showed that the *ESV* coefficients lacked elasticity to the value per unit area of each land use, and the research results were credible.

**Table 6.** The coefficient of sensitivity (*CS*) of Tianfu New Area from 2010 to 2020.

| Year | Farmland | Woodland | Grassland | Wetland | Water Body |
|------|----------|----------|-----------|---------|------------|
| 2010 | 0.3785 | 0.2127 | 0.0518 | 0.0011 | 0.3585 |
| 2020 | 0.2485 | 0.1835 | 0.0562 | 0.0019 | 0.5102 |

The *CS* of farmland was the highest in 2010 in Tianfu New Area because of the highest proportion of farmland, which had the greatest impact on *ESV*. In 2020, the water *CS* was the highest, due to the significant reduction of farmland area and the significant increase of water body area, combined with the performance of high *ESV* per unit area of water body. It indicates that the water body value coefficient and area change have the greatest influence on *ESV*.

### 4. Discussion

The concept of Park City embodies the theoretical and practical exploration of promoting ecological and sustainable urbanization in China. With its core concepts of ecological

civilization, it emphasizes people-oriented and green development [45]. Tianfu New Area is the "First Mentioned Place" and "First Experimental Demonstration District" of the concept of Park City. Since the beginning of planning, it has adhered to the concept of ecological priority and green development. This is the exploration and practice of China's Park City. Tianfu New Area, as a mega-city new area of Chengdu, presents its unique characteristics under the guidance of ecological civilization.

### 4.1. ESV Change and Its Driving Factors

From 2010 to 2020, due to a slight decrease in the correction coefficient $N_{i1}$, the *ESV* per unit area of farmland, woodland, and grassland, whose main service types are mainly affected by $N_{i1}$, showed varying degrees of decline, while the area of farmland and woodland decreased and grassland increased. The *ESV* of farmland and woodland decreased, while that of grassland increased. These results show that LULC change is the important driving force of *ESV*. In the study, the *ESV* per unit area and land use area of wetland and water body increased, so the *ESV* of them also increased. Especially, the area of water body increased significantly, and the value coefficient per unit area of water body was the highest, so the *ESV* of water body increased the most and contributed the most to the growth of urban *ESV*.

Tianfu New Area strictly implemented the ecological environment protection policy, strengthened the construction of water body and grasslands, and increased their land use areas in the process of orderly construction. In the 10 years, the urban ecological environment had been greatly improved, and the total *ESV* of the city had increased by 8.2%. This is consistent with the research results of other regions with similar research duration and geographical conditions in China. For example, Li et al. found that during 2008–2018, the *ESV* in the green heart area of Changsha–Zhuzhou–Xiangtan city group increased by CNY 0.88 billion, with an increase of 11.21% [46], and the growth trend is obvious. Li et al. believed that this was due to the orderly progress of urban construction and protection, and the urban construction mode combining rational development and strict control of the ecological bottom line could improve *ESV*. The reasons for the impact of land use changes on urban *ESV* in Tianfu New Area can be summarized as follows:

- The increase of area combined with the high-value coefficient per unit area of water body makes an outstanding contribution to the growth of urban *ESV*. In 2020, the area of the water body with the highest *ESV* per unit area in Tianfu New Area increased by 31.6% compared with that in 2010. The substantial increase of the water body area played a vital role in improving the total *ESV* of the city. It shows that in the urban planning and layout, under the condition that the total ecological space area is constant, it is essential to increase the area ratio of land types (such as water body, wetland, and woodland) with higher *ESV* per unit area to maintain or even improve the total *ESV* of the city.

- The increase of grassland and wetland areas promotes the promotion of urban *ESV*. From 2010 to 2020, the grassland and wetland areas in Tianfu New Area increased by 22.7% and 66.4%, respectively. The increase in grassland area comes from the former farmland, and the increase in wetland area comes from the former grassland and farmland. Both of them are transformed from land use with a low-value coefficient, thus effectively promoting the promotion of urban *ESV*.

- The reduction of farmland area reduces the expenditure on water supply services. In the 10 years, farmland was the land use type with the most obvious reduction in area. As the farmland has a negative value coefficient in water supply service, the reduction of the farmland area greatly reduces the negative effect of water supply, which plays an important role in the improvement of the urban *ESV*.

- Over the 10 years, due to the implementation of urban planning and eco-environmental protection policies, as well as the rapid development of population agglomeration and social economy, the value coefficients of water body and wetland in Tianfu New Area

had increased to a certain extent, and the *ESV* per unit area had increased, which had promoted the positive development of the urban *ESV*.

- In the process of urban development and construction, the conversion of farmland to construction land is the main type of land conversion. The *ESV* of farmland converted to construction land is decreasing. This part of the land conversion is the main area where *ESV* shows a decreasing state.

*4.2. Experiences and Suggestions on Urban Planning and Construction*

4.2.1. Ecological Planning Experiences of Tianfu New Area

- Planning Site Selection

The planning of Tianfu New Area incorporates the concept of ecological protection from the site selection. In September 2010, the Sichuan Provincial Party Committee and Government made a major strategic deployment to plan and construct the Tianfu New Area in Sichuan. Faced with the initial site selection for the construction of a new area in the northern plain area of Chengdu at that time, we practiced the concept of ecological priority and attempted to find new planning ideas to focus on protecting basic farmland. We provided theoretical support for the site selection of the new area from the perspective of ESs. Based on this, our proposal to leave the fertile land of the northern plain of Chengdu to future generations and plan and construct the new area in the southern and eastern hilly, mountainous, and plateau areas has been adopted by the Sichuan Provincial Party Committee and Government. We have been firmly practicing the concept of ecological civilization since the beginning of planning [18].

- Urban Scale

The determination of urban scale is the foundation of the master planning of the new area. When predicting population and construction land scale, Tianfu New Area takes resource constraints as the premise and starts from the perspective of resource-bearing capacity. Through the analysis of multiple factors such as land, ecology, and water resources, it identifies the short-board factor of bearing capacity and makes predictions based on careful consideration of the carrying capacity of each factor. Ultimately, the urban scale is controlled by the element of bearing capacity shortage (water resource factor), ensuring that the planning and construction scale of Tianfu New Area does not exceed the bearing capacity of the regional natural ecological environment [47].

- Land Layout

Tianfu New Area has abandoned the traditional urban construction model of "spreading big cakes" along the existing infrastructure. To avoid the problem of traditional "big city disease", Tianfu New Area adopts a "combined city" layout. At the same time, it adheres to the priority logic of ecological value protection and adopts the idea of "planning non-construction land first, and then construction land". Firstly, it identifies the ecological background within the region, reserves strategic "blank" areas such as ecological isolation zones, ventilation corridors, and urban green lungs between various "urban clusters", and provides rigid protection as non-construction land to enhance the resilience of cities to respond to natural disasters. Afterward, it searches for construction land space outside the scope of ecological protection, forming various "urban clusters". Ecological environment protection and ecological value enhancement enjoy priority in the planning process [48].

- Urban Management

In terms of urban management, Tianfu New Area has proposed a planning and management model that parallels expert consultation and administrative decision-making, as well as a "flat collaborative parallel" planning and management technology that is market-oriented and widely participated in by the masses. It institutionally guarantees the seriousness of the planning and management decisions, and provides technical support for the goal of "drawing the blueprint to the end" [49]. In the process of planning implementation, urban development has been scientifically organized according to the sequence

of planning before construction, and ecology before industry. At the same time, it vigorously promotes the whole city's large-scale afforestation actions and rapidly increases the urban and rural green space quantity and proportion [50]. Tianfu New Area has produced significant social, economic, and ecological benefits. The external evaluation states that: "The ecological environment of Tianfu New Area is very good, and achieving such results requires overall planning and unremitting efforts" [51]. The construction achievements of Tianfu New Area have laid the foundation for Chengdu to be approved as the only Park City demonstration area in China.

### 4.2.2. Suggestions for Ecological Space Planning

In terms of ESs supply, the level of urban *ESV* first depends on the proportion distribution of ecological land and urban construction land, and second on the area distribution and spatial distribution of different land use types within ecological land. In addition, differences in production capacity of the same land use type are also essential influencing factors. Through the study of land use and *ESV* in Tianfu New Area, it was found that there are problems such as the imbalance in the proportion of land use types and the need to continue improving the quality of land use. The proportion of land use types such as water bodies, wetlands, and woodlands with high *ESV* per unit area is relatively low, which affects the efficient expression of the overall *ESV*. From the perspective of improving the *ESV*, subsequent ecological space planning and construction should be based on terrain and current ecological elements (such as the distribution of mountains and water systems), systematically planning the spatial layout of woodlands, water bodies, wetlands, etc., adjusting the proportion of different types of ecological land. It should vigorously plant trees and forests in ecologically sensitive and vulnerable areas such as mountains, steep slopes, water conservation, and soil erosion, and transform them into landscape types with high *ESV* per unit area such as woodlands, wetlands, and water bodies.

- During the implementation of the planning, strict protection should be exercised over large blue and green "source" patches such as urban ventilation corridors to prevent urban construction from sticking to the main urban area of Chengdu. Ecological space planning should be integrated into the overall ecological network pattern of Chengdu.
- Taking Longquan Mountain as the source of forests, Tianfu New Area should continue to strengthen the vegetation coverage of Longquan Mountain, increase the proportion of local broad-leaved tree species, and gradually transform existing scattered farmland and economic forest land developed by farmers into urban forests, highlighting Longquan Mountain's functional status as the "green lung" of Tianfu New Area.
- There is a need to increase the area of water bodies and wetlands based on the current trend of the water network; to plan strip green spaces along the water system and major urban roads to form a "corridor" connecting large green patches, enhancing the overall connectivity of ecological space; to establish urban wetlands in sensitive areas such as water conservation, and strictly protect them to prevent water system encroachment and pollution.
- The main terrain in Tianfu New Area is shallow hilly terrain. The current land use method of combining some shallow hills with cultivated land has led to significant soil erosion in the area. It is recommended to combine the spatial distribution map of soil erosion intensity and implement strict afforestation and ecological protection policies in the Longquan Mountain area with steep slopes and severe soil erosion, as well as in the southern hilly and terraced areas, to prevent soil erosion and improve vegetation coverage.

### 4.3. Limitations and Future Research

First of all, the planning and construction of Tianfu New Area started in 2010 and then entered the stage of rapid urbanization. Before that, LULC changed slowly. In view of the special situation of Tianfu New Area as a mega-city new area, it is of typical significance to study the dynamic changes of LULC and urban *ESV* during rapid construction and

development over 10 years. Therefore, the research time interval of this study is limited, so we can continue to pay attention to the dynamic changes in land use and urban *ESV* in the future.

Secondly, the evaluation of *ESV* in this study takes urban ecological space as the research object. Due to the significant difference between urban construction land and urban ecological space in *ESV* contribution, the *ESV* of urban construction land is not considered in the research. In the future, we can pay attention to the role of urban construction land in *ESV* on the basis of this study and carry out special research to explore its impact mechanism in urban *ESV*.

Thirdly, taking into account the temporal dynamics and spatial heterogeneity of the *ESV* value coefficient, the study revised the *ESV* value coefficient in the EFM by Gaodi Xie et al. [5,12] both from the aspects of natural geography and social economy. The correction is based on the previous work by Li and Qiu [18] and the specific natural geography and socio-economic conditions of Tianfu New Area. Due to different social and economic development stages and different natural environment conditions in different countries and regions, *ESV* evaluation results are affected [52,53]. Regarding the research approach proposed in this study, it may have the potential to be applied in other contexts, in which the research regions can choose representative correction factors to revise the *ESV* value coefficient of space–time heterogeneity.

## 5. Conclusions

Based on the analysis of land use and *ESV* changes from 2010 to 2020, this study reveals the ecological effects of rapid urbanization in Tianfu New Area. In terms of land use, the reduced farmland was mainly converted into construction land. In addition to construction land, farmland conversion to grassland and farmland conversion to water body are the main types of land conversion. The area of water body increased significantly. The land use dynamic index (K) of water body and grassland were 3.16% and 2.27%, respectively, showing a significant growth trend. The results show that the total *ESV* of Tianfu New Area increased from CNY 8711.63 million in 2010 to CNY 9426.62 million in 2020, with an increase of CNY 714.99 million. Due to its high-value coefficient per unit area, water body surpassed farmland and became the largest contributor to urban *ESV* in 2020. On this basis, we calculated the elasticity of *ESV* due to LULC changes and found that a 1% change in LULC led to an average 0.25% change in total *ESV*, which is similar to the conclusion of former studies. In a theoretical aspect, this study selected correction factors from natural geography and social economy aspects and established a comprehensive dynamic evaluation model for the ecosystem service value of specific urban areas. Based on the evaluation model above, we reveal the ecological effects of rapid urbanization in Tianfu New Area. In a practical aspect, we have summarized the ecological planning and construction experiences of Tianfu New Area from planning site selection to urban management. Moreover, we propose suggestions for ecological space planning of Tianfu New Area to further improve the quality of the ecological environment. The results show that the development and construction of new urban areas do not necessarily lead to the inevitable decline of *ESV*. The rational land use layout under the guidance of the scientific planning concept, combined with strict urban management and environmental protection policies, will help to achieve the dual effects of development and protection, which has important implications for the development and construction of urbanized areas and ecological protection.

**Author Contributions:** Conceptualization, J.L. and J.Q.; Data curation, J.L.; Formal analysis, J.L.; Funding acquisition, J.Q.; Investigation, J.L.; Methodology, J.L.; Project administration, J.L.; Resources, J.L. and J.Q.; Software, J.C.; Supervision, J.Q.; Validation, J.L., M.A.-B., Y.W. and M.Y.; Visualization, M.A.-B. and Y.W.; Writing—original draft, J.L.; Writing—review and editing, J.L., J.Q., M.A.-B., Y.W., M.Y. and J.C. All authors have read and agreed to the published version of the manuscript.

**Funding:** This research was funded by the National Natural Science Foundation, grant number 52078423, the Science and Technology Support Plan project of Sichuan Province, grant number 2020YFS0054, and the Sichuan Provincial Science and Technology Innovation Base (Platform) and Talent Plan, grant number 2022JDR0356.

**Data Availability Statement:** The data presented in this study are contained within this article.

**Conflicts of Interest:** The authors declare no conflict of interest.

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
