# Peer review of "A Modified Equivalent Factor Method Evaluation Model Based on Land Use Changes in Tianfu New Area"

_land, doi:10.3390/land12071335_

Round 1

Reviewer 1 Report

The ESV of urban construction land is not considered in the research. It should be further clarified why this choice does not affect the results. Is it truly possible that urban transformation does not negatively offset the benefits derived from the increase in water bodies?

It is essential to provide a more detailed explanation of this aspect.

Author Response

Dear reviewer,

We are pleased to note the favorable comments of the reviewers. The manuscript has been modified according to the comments. The detailed corrections are as follows:

Responses: Previous studies generally concluded that construction land does not have ecosystem service value[1-2], but recent studies found that construction land also had ecological functions such as culture and recreation[3-4]. According to Maimaiti’s research[4], the ecosystem service value per unit area of construction land is 0.01 times that of farmland. Farmland is the land use type with the lowest ecosystem service value per unit area in Tianfu New Area in our study. Due to the essential differences in ESV contributions between urban construction land (excluding parks and green spaces) and urban ecological spaces, the ecosystem service value of urban construction land was not considered in our study. In the future, based on this article, special research can be conducted on the interaction between urban construction land and urban ecological space, exploring its impact mechanism in urban ESV(4.3 Limitations and Future Research). The newly added urban construction land in Tianfu New Area mainly comes from farmland. Due to the planning and construction of the Tianfu New Area in the hilly and mountainous areas in the southern and eastern parts of Chengdu, the service value of the farmland in this area is lower than that of ordinary plain areas. (1) In terms of the value of ecosystem services per unit area, in 2020 of Tianfu New Area, water bodies were 42 times higher than farmland, wetlands were 15 times higher, and grasslands were 3 times higher. The areas of water bodies, wetlands, and grasslands were all showing an increasing trend. (2) Due to the negative value coefficient of farmland in terms of water supply service, the reduction of farmland area significantly reduces the negative effect of farmland water supply service, which has an important promoting effect on the improvement of urban total ESV. (3) The calculation of urban ESV in this paper considers the impact of scarcity of resources and people’s ability to pay on ESV. Due to the scarcity of ecological resources caused by population agglomeration, combined with the increase of per capita income brought about by the rapid development of social economy, the scarcity of ecological resources and the people’s ability to pay have been improved. The value correction coefficients of water bodies, wetlands and other areas have increased to a certain extent, and the ESV per unit area has increased, which has promoted the positive improvement of the overall ESV of the city (4.1 ESV Change and Its Driving Factors).

Reviewer 2 Report

I suggest you use only km2 instead of hm2 (ha) as surface unit

And that you use the higher sign 100’000.00  as thousands separator instead of writing  100,000,00

5. Conclusions

In terms of land use, the area of farmland decreased by 25,752.7 hm2, the area of construction land increased by 23,353.7 hm2

I think the use of comma and dot sign are confusing.

Better would be

In terms of land use, the area of farmland decreased by 25’752.7 hm2, the area of construction land increased by 23’353.7 hm2 (if that is what you mean.

Even better would be

257,527 km2 .. 233,537 km2 or 257,53 … 233,54 km2  as it read faster

Same goes for
Table 1.
Table 2.

In 3.3.2. “0.5-1 million yuan/hm2” would be “50-100 million yuan/km2”

increase of 974,513.7 yuan/hm2 (increase of 974’513.7 yuan/hm2) would be “97 million yuan/km2”

etc. , etc.

Please check all numbers again with the new units ..

Author Response

Dear reviewer,

Thank you for your favorable and constructive comments. We have carefully reviewed and addressed your suggestions. As per your feedback, we have made sure to update all the numerical values in the manuscript with the appropriate units. These changes are now reflected consistently throughout the entire document.

Thank you once again for your valuable comments.

Reviewer 3 Report

The paper proposes a Modified Equivalent Factor Method (MEFM) to evaluate Ecosystem Service Value (ESV) and explores how Land Use and Land Cover Changes (LULCC) impact ESV in the process of rapid urbanization of Tianfu New Area from 2010 to 2020. This research topic focusing on the interaction between urbanization and ecosystem services is of important scientific significance and practical application value. The research methods are systematic and the results are instructive. Overall, this draft paper has the potential to be high-quality research work after some revisions:

Title

1: The title is relatively concise. I would suggest modifying it to "A Modified Equivalent Factor Method Evaluation Model Based on Land Use Changes in Tianfu New Area" or something similar. The modified title can more clearly and accurately express the content of the study.

Abstract

2: There are some language expression issues in lines 10-11 of the abstract, "build-10ing" should be modified to "building", and "equiv-11alent" should be modified to "equivalent".

3: The sentence expression in lines 13-15 of the abstract is rather bloated. I would suggest modifying it to: "This study selected correction factors from natural geography and social economy aspects, and established spatiotemporal correction models for standard equivalent coefficients as well as a comprehensive dynamic evaluation model for the ecosystem service value of specific urban areas." The modified expression is more standard and fluent.

4: The sentence expressions in lines 17-19 and 21-23 of the abstract are also relatively bloated. I would suggest the following modifications:

Lines 17-19: "The area of farmland decreased the most. The areas of construction land, grassland, and water bodies increased significantly. The reduced farmland was mainly converted into construction land, followed by grassland and water bodies. Other land use types had smaller changes.” 

Lines 21-23: “Due to the increased area of water bodies and their high value coefficient per unit area, the urban ecosystem service value showed an increasing trend. During the study period, the conversion of about 1% of land led to about a 0.25% change in the urban ecosystem service value.”

5: In the last sentence of the abstract, "new urban areas" should be modified to "newly built urban areas" or "newly developed urban areas" to avoid ambiguity.

Introduction:

1: The first two sentences in Paragraph 3 are somewhat repetitive. I would suggest combining them into: “Relying on ecological economics and environmental economics, the value evaluation method regards ESs as valuable goods. It directly reflects the total value and scarcity level of ESs from the perspective of monetary value[7].”

2: In Paragraph 4, the second sentence "hugely" should be "hugely" to make the expression more objective and accurate.

3: The last sentence of Paragraph 5 is too long. I would suggest dividing it into two sentences to improve readability: “The equivalent value per unit area of ES in China[5,12] expresses the static value equivalent of ESs at a national scale and a specific time, ignoring the spatial heterogeneity and the dynamic changes over time caused by regional differences in biomass within the same land use type. In addition, Xie Gaodi’s research objects mainly focus on natural ecological space, and ESV is mainly affected by natural geographical factors.”

4: The first sentence of Paragraph 6 can be refined to: "According to the practical needs of the planning and construction of Tianfu New Area, Li and Qiu proposed value correction models based on systematic literature review and expert interviews."

5: In Paragraph 10, "How to" at the beginning of the first sentence can be deleted to express more concisely. "Is a long-standing challenge[15,18]" in the second sentence can be moved to the end of the first sentence to make the logic clearer.

6: The last sentence of Paragraph 12 "It can promote the improvement of urban space and living environment quality, which is conducive to the transformation of cities into green sustainable development[21]." is too long. I would suggest dividing it into two sentences: "It can promote the improvement of urban space and living environment quality. This is conducive to the transformation of cities into green sustainable development."

7: The last two sentences about research significance in Paragraph 15 can be refined to:  "Our research focuses on the impact of LULC changes on urban ESV of the mega-city new area in the context of China’s national “Park City”. The research significance is to provide a reference for ESV evaluation and ecological space planning in urbanized areas, as well as suggestions for urban ecological policy making.”

Discussion:

1: The last sentence of Paragraph 1 "Since the beginning of planning, it has adhered to the concept of ecological priority and green development, which is the exploration and practice of China's Park City.” is too long. I would suggest dividing it into two sentences: " Since the beginning of planning, it has adhered to the concept of ecological priority and green development. This is the exploration and practice of China's Park City. "

2: In Paragraph 2, the first sentence "From 2010 to 2020, the ESV per unit area of farmland, woodland and grassland in Tianfu New Area decreased to different degrees." Why did it decrease? This result differs from Expectation. I would suggest the author provide an explanation here. 

3: Paragraph 3 mentions "The reasons for the impact of land use changes on urban ESV upgrading in Tianfu New Area can be summarized as follows:", but in fact, it only lists the reasons for the increase in ESV, without mentioning the reasons for the decrease. I would suggest the author supplement the reasons for the decrease in ESV in this paragraph to comprehensively summarize the impact of LAND USE changes on ESV.

4: The title of Section 4.1 "Impact of Land Use Changes on ESV" can be more concise. I would suggest changing it to "ESV Change and Its Driving Factors". This subtitle can better summarize the content of this section, that is, how LAND USE changes drive changes in ESV.

5: The title of Section 4.2.1 "Experiences on Planning and Construction of Tianfu New Area" is rather stiff in expression. I would suggest changing it to "Ecological Planning Experiences of Tianfu New Area". This title can more clearly express the focus of this subsection on the ecological planning experience of the new area. 

6: The title of Section 4.2.2 "Strategies and Suggestions for Improving ESV in Tianfu New Area" is logically abrupt. I would suggest changing it to "Suggestions for Optimizing Ecological Network" or "Suggestions for Ecological Space Planning" to connect more coherently with the previous subsection "Ecological Planning Experiences" logically. The suggestion can correspond to specific recommendations related to the ecological network and space planning.

7: The content from Paragraph 5 to Paragraph 7 on planning positioning, urban scale and land layout is described in detail, but the content on urban management only mentions general principles, lacking discussion on specific management measures and their implementation effects. I would suggest the author supplement the specific measures and effects of urban management here to enable the experience achieved by the new area in all aspects of ecological planning to be expressed more comprehensively and systematically.

Conclusions

1: The first sentence "This study analyzes the characteristics of land use and urban ESV changes in the rapid development of Tianfu New Area from 2010 to 2020." is too general as the opening sentence of the Conclusions section. I would suggest changing it to: "Based on the analysis of land use and ESV changes from 2010 to 2020, this study reveals the ecological effects of rapid urbanization in Tianfu New Area." This sentence can more clearly express the research content and findings.

2: The second sentence "In terms of land use, the area of farmland decreased by 25,752.7 hm2, the area of construction land increased by 23,353.7 hm2, and more than 80 % of the reduced farmland was converted into construction land." is too detailed and does not need to be repeated in the Conclusions section. I would suggest deleting this sentence.

3: The fourth sentence "The ESV evaluation of this study inherits the advantages of the EFM, and comprehensively considers the regional differences in natural geography and social economy according to the characteristics of urbanization areas." is repetitive in the Conclusions section. I would suggest deleting it.

4: The sixth sentence "Due to the increase in area and the high-value coefficient per unit area, water body surpassed farmland and became the first contributor to the urban ESV in 2020." is too long. I would suggest dividing it into two sentences: "The area of water body increased significantly. Due to its high value coefficient per unit area, water body surpassed farmland and became the largest contributor to urban ESV in 2020.”

5: The eighth sentence "The reliability of our evaluation of ESV was tested by sensitivity analysis in the research." does not need to be repeated in the Conclusions section. I would suggest deleting this sentence.

6: The ninth sentence "In all cases, the CS is less than 1, and the evaluation result is reliable, which is similar to the results of He[31] and Zhu[15]." is also repetitive in the Conclusions section. I would suggest deleting it.

7: The last three sentences on the research significance are rather general and abstract. I would suggest that the author further clarify the specific theoretical and practical contributions of his research here to give readers a clearer understanding of the scientific goals and social impacts that the author intends to achieve.

Please reduce the length and improve conciseness

Round 2

Reviewer 3 Report

The quality of the modified version has been improved to some extent.

Minor editing of English language required